# Ferulic Acid reduces amyloid beta mediated neuroinflammation through modulation of Nurr1 expression in microglial cells

**Ali Moghimi-Khorasgani[1,2], Farshad Homayouni Moghadam [2]\*, Mohammad Hossein Nasr-Esfahani [2]**

**1** Department of Biology, Faculty of Science and Technology, ACECR Institute of Higher Education (Isfahan Branch), Isfahan, Iran, **2** Department of Animal Biotechnology, Cell Science Research Center, Royan Institute for Biotechnology, ACECR, Isfahan, Iran

\* homayouni@royan-rc.ac.ir

## Abstract

Microglial cells (MGCs) serve as the resident macrophages in the brain and spinal cord, acting as the first line of immune defense against pathological changes. With various phenotypes, they can shift from a homeostatic state to a reactive state or transit from a reactive to a non-inflammatory reactive state (alternative homeostatic). A well-timed transit is crucial in limiting excessive microglial reaction and promoting the healing process. Studies indicate that increased Nurr1 expression promotes anti-neuroinflammatory responses in the brain. In this study, we investigated the possible role of ferulic acid (FA) in facilitating microglia transition due to its anti-inflammatory and Nurr1-inducing effects. MGCs were extracted from the brains of male NMRI mice at postnatal day 2 (P2) and cultured with or without FA and beta-amyloid (Aβ). Real-time qRT-PCR was conducted to measure the expressions of *Nurr1*, *IL-1β*, and *IL-10* genes. Immunostaining was performed to determine the number of NURR1-positive cells, and the ramification index (RI) of MGCs was calculated using Image J software. Treating MGCs with FA (50 μg/ml) induced *Nurr1* and *IL-10* expressions, while reducing the level of *IL-1β* in the absence of Aβ-stress. Further assessments on cells under Aβ-stress showed that FA treatment restored the *IL-10* and *Nurr1* levels, increased the RI of cells, and the number of NURR1-positive cells. Morphological assessments and measurements of the RI revealed that FA treatment reversed amoeboid and rod-like cells to a ramified state, which is specific morphology for non-inflammatory reactive microglia. To conclude, FA can provide potential alternative homeostatic transition in Aβ-reactive microglia by recruiting the NURR1 dependent anti-inflammatory responses. This makes it a promising therapeutic candidate for suppressing Aβ-induced neuroinflammatory responses in MGCs. Furthermore, given that FA has the ability to increase NURR1 levels in homeostatic microglia, it could be utilized as a preventative medication.

**Data Availability Statement:** All relevant data are within the paper and its Supporting Information files.

**Funding:** The author(s) received no specific funding for this work.

**Competing interests:** The authors have declared that no competing interests exist.

**Abbreviations:** FA, Ferulic Acid; MGCs, Microglial cells; P2, postnatal day 2; Aβ, beta amyloid; RI, Ramification Index; AD, Alzheimer's disease; IL, Interleukin; TNF-α, tumor necrosis factor-α; NO, nitric oxide; ROS, reactive oxygen species; LPS, lipopolysaccharide.

# Introduction

Ferulic acid (FA, hydroxycinnamic acid) is a natural phenolic compound and is abundant in the leaves and seeds of many plants, predominantly brown rice, wheat, and barley [1]. FA has recently been considered as an anti-aging chemical because of its neuroprotective effects [2–4]. Regarding its effects on neurons, many medicinal properties have been attributed to FA, including: proliferative [5], anti-inflammatory [6, 7], and anti-oxidative properties [7]. Also, its supportive function has been reported on the improvement of various nervous system disorders, including brain ischemia [8], spinal cord injury [9], and Alzheimer's disease (AD) [10, 11].

Research on microglial cells has revealed that FA exhibits potent anti-oxidative and anti-inflammatory properties, effectively reducing the release of pro-inflammatory cytokines while increasing the expression of anti-inflammatory cytokines [12]. *In vivo* studies have shown that treatment with FA reduces inflammation by reducing IL-1β expression in the models of neurodegenerative disorders [7, 13, 14].

It has been widely reported that FA prevents beta-amyloid (Aβ) aggregation, destabilizes Aβ fibrils, and protects neurons against Aβ-induced toxicity [15–18]. Aβ acts as a neuro-inflammatory factor and stimulates the shift of homeostatic microglia toward reactive microglia, which is recognized as one of the contributing factors in the pathogenesis and progression of AD [19–21]. Recent studies revealed that reactive microglial cells could also trigger the transition in astrocytes toward reactive astrocytes (A1) [22].

Reactive microglia induced by aging, Aβ and metabolic stress produce pro-inflammatory cytokines (e.g., IL-1β, IL-6, TNF-α) and free radicals (e.g., NO, ROS), which can directly and indirectly contribute to neuronal loss in AD. [21–24]. Activation of anti-inflammatory pathways may offer a means of mitigating the harmful effects of reactive microglia-induced inflammation [25]. One of the transcription factors that can inhibit inflammation caused by Aβ is Nurr1 [26, 27], whose structural defects or reduced expression have been shown to be associated with increased amyloid plaque deposition [26]. As a transcription factor, Nurr1 plays a crucial role in the development of dopaminergic neurons and contributes to the activation of neuroprotective and anti-inflammatory pathways [28]. Dysfunction or reduced expression of Nurr1 has been associated with impaired energy metabolism, mitochondrial dysfunction, and oxidative stress [29, 30]. Given the association between reduced Nurr1 expression and these changes in dopaminergic neurons, beside anti-synucleinopathy therapy, increasing Nurr1 expression has been suggested as a promising therapeutic target for Parkinson disease [28]. However, fewer studies have been conducted on its role in microglial and astrocytic inflammations.

Recent studies report that the increased expression of Nurr1 in microglia and astrocytes can reduce the expression of RasGRP1 and thereby play a key role as an anti-inflammatory mediator in reducing neuroinflammation [31]. Nurr1 exerts its anti-inflammatory effects by binding to NF-kappaB-p65 on the promoters of inflammatory genes in a signal-dependent manner. In this way, Nurr1 activates the CoREST corepressor, eliminates the effect of NF-kappaB-p65, and suppresses the transcription of inflammatory genes [32]. According to Oh et al., Nurr1 can act as an anti-neuroinflammatory mediator by negatively regulating RasGRP1 expression at the transcriptional level, as observed in inflammatory conditions induced by lipopolysaccharide (LPS) [31].

Some studies suggest that LPS exposure increases Nurr1 expression in microglia and astrocytes [32, 33]. However, Nurr1 regulation is not straightforward. Lallier et al. (2016) found that while whole-brain homogenates show an increase in Nurr1 expression in response to inflammation, primary microglia and BV2 cells exhibit a decrease in Nurr1 expression under

similar conditions. These findings suggest that Nurr1 expression may increase in other cell types rather than microglia [26]. All these studies jointly confirm the role of Nurr1 in inflammation, but whether its expression is more important in astrocytes or microglial cells is still a research question. There is also evidence that there is a decrease in the number of glutamatergic neurons expressing Nurr1 in AD subjects, indicating that Nurr1 may play an important role in the inflammatory responses seemed in some other types of neurons [34].

To date, no studies have investigated the effect of FA on Nurr1 expression in primary microglial cells. However, the expression of Nurr1 increases in neurons treated with FA, which leads to the expression of dopaminergic differentiation markers [35]. According to the findings proving the anti-inflammatory effect of FA and the role of Nurr1 in neuroinflammation, the ability of FA to influence the expression of Nurr1 in mouse primary microglia was investigated in the present study.

We used primary mice microglia as recent studies have shown that immortal microglia cell lines are both genetically and functionally distinct from primary microglia [36]. The presence of microglial states observed in mice and their existence in humans are still a topic of debate. It is essential to validate and translate these findings across species to prevent clinical trial failures resulting from limitations of animal models [37]. Furthermore, since human primary microglia are not easily accessible for research, studying primary murine cell lines could be advantageous in identifying possible similarities and differences.

## Materials and methods

### Ethics statements

All animal experiments were approved by the Institutional Review Board and Institutional Ethical Committee of the Royan Institute (IR.ACECR.ROYAN.REC.1399.033).

### Media and chemicals

All media and chemical reagents were obtained from Thermofisher (Gibco, MA, USA) company unless otherwise specified.

### Animals

Male neonate NMRI mice (n = 3–4 for each experiment in a 6-well culture plate, age = 2 days' post neonatal (P2)) were used for cell isolation. Animals were kept in the light/dark cycle (12/12 h), neonates were kept in cages with their mother and had full access to breastfeeding, and the mother had easy access to water and food. Care and maintenance of mice were carried out according to the approved regulations for studies using laboratory animals by the Royan Institute.

### Brain extraction and isolation of mixed glial cells

Microglia were isolated from mixed glia obtained from the brains of male neonate mice. According to the Guidelines of Ethics Committee of Royan Institute for the Euthanasia of Animals which were in accordance with Iranian National Research Ethics Committee hypothermic anesthesia was used to alleviate the suffering of mouse pups before euthanizing them via decapitation with a razor blade [38]. Briefly, brains were excluded from the head under sterile conditions. After washing with PBS containing 2% penicillin/streptomycin, they were sliced by a surgical blade, and slices were incubated with trypsin-EDTA 0.05% (Cat. No. 25300054) for 20 minutes. Then enzyme was neutralized using the culture medium (indicated below), and afterward the medium containing cells (mixed glia) was passed through the nylon mesh

(70 microns) to separate the debris. Next, mixed glia was collected by centrifugation and transferred to T75 flasks in the culture medium consisting of: Dulbecco's Modified Eagle's medium (DMEM, Cat. No. 11960044) + 10% fetal bovine serum (FBS, Cat. No. 12483020) + 1% Penicillin/Streptomycin (Cat. No. 10378016) + 1% NEEA (Cat. No. 11140050) + 1% Glutamax (Cat. No. 35050061), and incubated in cell culture incubator at 37˚ C and 5% $CO_2$ [39].

## Microglial cell isolation from mixed glia

The mixed glia culture continued until they reached nearly 90% of confluency. In this part, to isolate microglial cells from mixed glia, their culture medium was replaced with a fresh one, and flasks were placed for one hour in the shaking incubator (37˚C and 120 rpm) to detach microglia from underlying cells mechanically. The supernatant containing microglia was transferred into the centrifuge tube and collected by centrifugation, and after counting, they were cultivated in a 6-well plate coated with Poly-L-Lysine. The culture plates were already coated with 0.01% Poly-L-Lysine (Sigma-Aldrich, MA, USA, Cat. No. P4707) overnight at 37˚C. Cell counts were performed using a hemocytometer chamber slide, and cell counts were performed for cells extracted from each animal/flask [39]. Typical yield by this method is around 150000–200000 microglia/brain.

## Experimental groups

MGCs were plated in Poly-L-Lysine coated 6-well culture plates ($10^5$ cells/well) for following treatments with FA (trans-Ferulic acid, Sigma-Aldrich, MA, USA, Cat. No. 128708). FA treatments were performed at 0, 50, 100, 250, and 500 μg/ml concentrations. Twenty-four hours later, the relative expressions of Nurr1, IL-1β, and IL-10 genes were measured using RT-qPCR.

After finding the most effective concentration of FA on Nurr1 expression (50 μg/ml), that was used for further experiments to assess the effect of FA in the presence of beta-amyloid induced inflammation (Fig 1). The treatment groups studied in this section were: **1)** control group (Ctrl), **2)** FA 50μg/ml (FA) (24 hours), **3)** Aβ 50 nM (48 hours), and **4)** Aβ 50 nM for 24 hours followed by FA 50 μg/ml (Aβ+FA) for another 24 hours. Then gene expression assay, RI measurement, and cell staining assay were performed to find the differences between groups. All of the experiments were done in three repeats and replicates. The RI and morphological types of MGCs were also determined, as described below.

## Ramification index (RI) and cells morphology assessments

The RI and morphological phenotypes of MGCs were determined by analysis of cell images. For this purpose, ten non-overlapping fields were captured from each well of a 6-well culture plate, then the RI of cells was determined using Scholl analysis assay by Image J software [40]. Also, the number of rod-like, ramified, and amoeboid cells were also counted in each image using image j software [41].

## Immunocytochemistry

Immunostaining was performed against the Nurr1 protein to determine the percentage of Nurr1 positive MGCs [28]. Cultured samples were washed three times with PBS and fixed in 4% paraformaldehyde for 10 min at RT. Then, permeabilization was performed using incubation with 0.1% Triton X-100 for 10 min. Nonspecific antibody binding was blocked by incubation with 1% BSA in PBS for 30 min at 37˚C. Samples were incubated with the appropriate primary antibody, namely, Polyclonal anti Nurr1 antibody produced in Rabbit (Sigma-Aldrich, MA, USA, Cat. No. sc-5568), at 4˚C overnight. Then they were washed five times

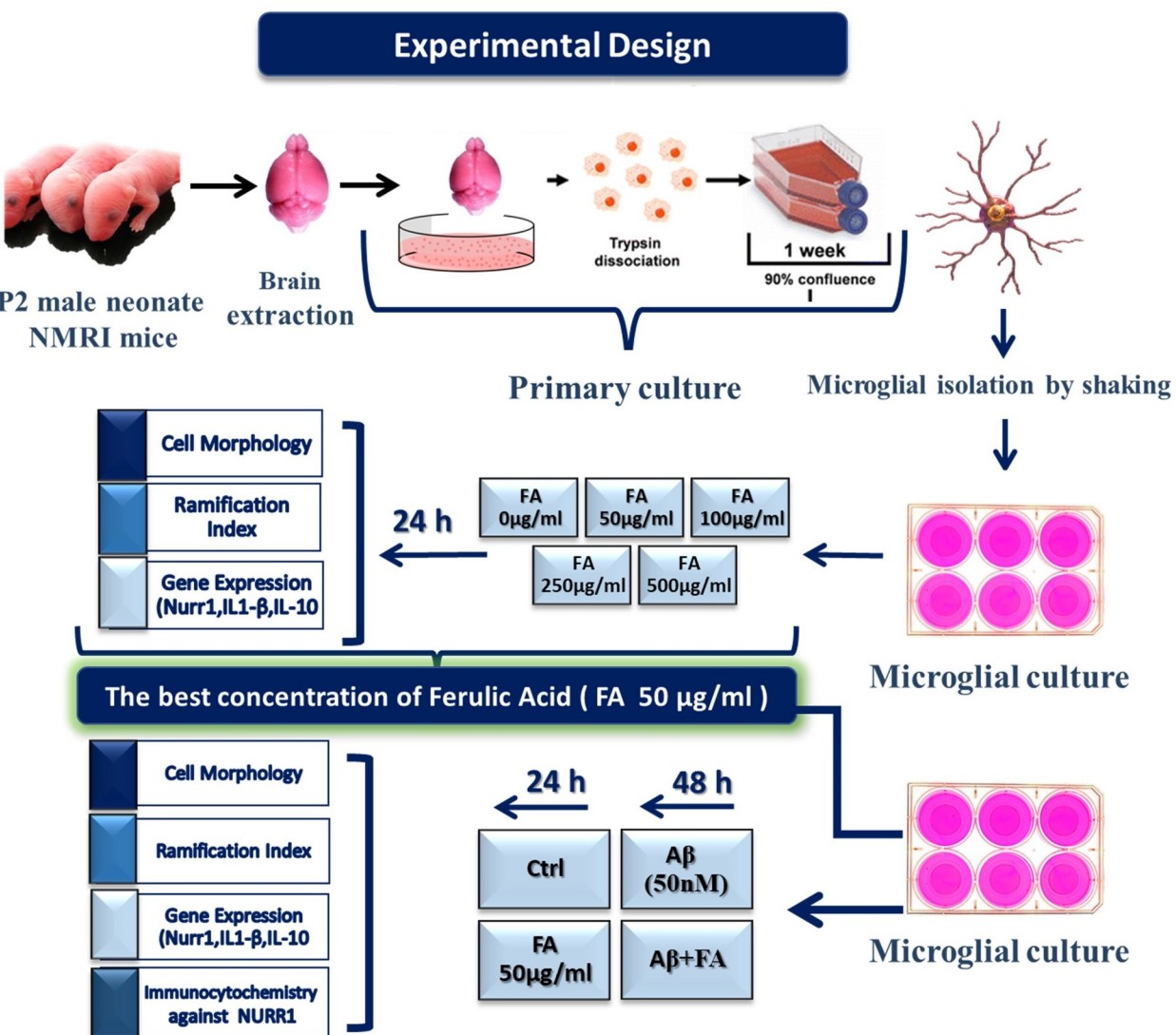

**Fig 1. Experimental design of the present study.** Microglial cells were isolated from neonate mice and at the first step they were treated with different concentrations of ferulic acid (FA) to find the most effective concentration on *Nurr1* induction. Then in the second round of experiments cells were treated with FA in the presence of beta-amyloid (Aβ) stress to find the response of cells after exposure to the inflammatory signal.

with PBS/0.1% Tween-20 for 5 min/each and incubated with the FITC conjugated secondary antibody (Mouse Anti-Rabbit IgG, Sigma Aldrich, MA, USA, Cat. No. sc-2357), in PBS for 1 hour at RT. After three times washes with PBS+0.1% Tween-20 for 10 min, cells were counter-stained with DAPI (Sigma-Aldrich, MA, USA, Cat. No. D8417) [42]. To determine the number of Nurr1 positive cells, ten images were captured from random areas of each well and Nurr1 and DAPI-stained cells (nuclei) were counted separately for each image.

## RNA extraction and real-time PCR

Total RNA was extracted from cells using RNase mini kit (Qiagen, Hilden, Germany, Cat. No. 74106) according to the manufacturer's instructions. RNA concentration was assessed by measuring the absorbance at 260 nm. Two micrograms of RNA were used for cDNA synthesis using random hexamer primer mix via first strand cDNA synthesis kit according to the

**Table 1. Primer sequence.**

| Gene name | Forward primer | Reverse primer |
|---|---|---|
| Gapdh | 5-TGCCGCCTGGAGAAACC-3 | 5-TGAAGTCGCAGGAGACAACC-3 |
| Nurr1 | 5-TGGCTATGGTCACAGAGA-3 | 5-GTAGTTGGGTCGGTTCAA-3 |
| IL-1β | 5`-ATTAGACAACTGCACTACAGG-3` | 5`-ACAGGTATTTTGTCGTTGCTTG-3` |
| IL-10 | 5`-ATTTTAATAAGCTCCAAGACCA-3 | 5`-GTCCAGCAGACTCAATACACA-3` |

manufacturer's instructions (Takara, Shiga, Japan, Cat. No. RR037A). Real-time reverse transcription PCR (qRT-PCR) reactions were performed in the Rotor Gene Q Real-Time PCR cycler (Qiagen, Hilden, Germany). For each reaction, the synthesized cDNA (40 ng) was subjected to PCR by mixing with 10 μL of SYBR Green master mix (SYBR® Premix Ex Taq™ II, Takara, Shiga, Japan, Cat. No. RR820L), 0.5 μM of each primer (Table 1) in a total volume of 20 μL at the annealing temperature mentioned in Table 1. The 2-ΔΔCT method was used to calculate the relative quantification of gene expressions. The primer sequences, annealing temperature, and product size are listed in Table 1.

## Data analysis

Data are presented as mean ± SEM, GraphPad Prism v8 software was used for statistical analysis. The significant differences were determined among groups using one-way ANOVA test, and p≤0.05 was assumed significant. Significant differences were shown in figures using *, **, ***, **** symbols which respectively represent *P < 0.05, P < 0.01, P < 0.001*, and *P < 0.0001*.

## Results

### FA treatment increased *Nurr1* and *IL-10* expressions in primary MGCs while decreased *IL1-β*

In this section, to evaluate the effect of FA on cells in normal conditions (without the presence of inflammatory stress, homeostatic state), primary MGCs were treated with different concentrations of FA (50, 100, 250, and 500 μg/ml) and 50 μg/ml was the most effective concentration in the induction of changes in morphology and genes expressions (Fig 2).

As shown in Fig 2A & 2B, the number of ramified microglia has increased after treatment with FA 50 and 100 μg/ml, but 50 μg/ml exerted the significant change compared to the control group (*P < 0.0001*) while it was not significant in the 100 μg/ml group. As the means of RI were 3.2 ± 0.18, 4.8 ± 0.34, and 3.7 ± 0.16 in the control, 50 and 100 μg/ml groups respectively. While in the 250 and 500 μg/ml groups, the means of RI were significantly decreased compared to the control group *P < 0.05* and *P < 0.0001* respectively.

As shown in Fig 2Ca, when comparing the differences in gene expressions the *Nurr1* expressions, were significantly increased in 50 (6.27-fold), 100 (6.06-fold) and 250 (4.17-fold) groups as compared to the control group (*P < 0.0001*). While expressions of *IL1-β* (Fig 2Cb) were significantly decreased in 50 (0.56-fold), 100 (0.65-fold) and 250 (0.78-fold) groups compared to the control group (*P < 0.0001, P < 0.0001* and *P < 0.01*, respectively). While it significantly (*P < 0.0001*) increased (1.74-fold) in the 500 μg/ml group as compared with the control group.

The expression rate of another anti-inflammatory cytokine *IL-10* represented that (Fig 2Cc) it was increased significantly in 50, 100 and 250 μg/ml groups compared to the control group (*P < 0.0001, P < 0.0001* and *P < 0.001*, respectively) while treatment with 500 μg/ml had no significant effect on its expression.

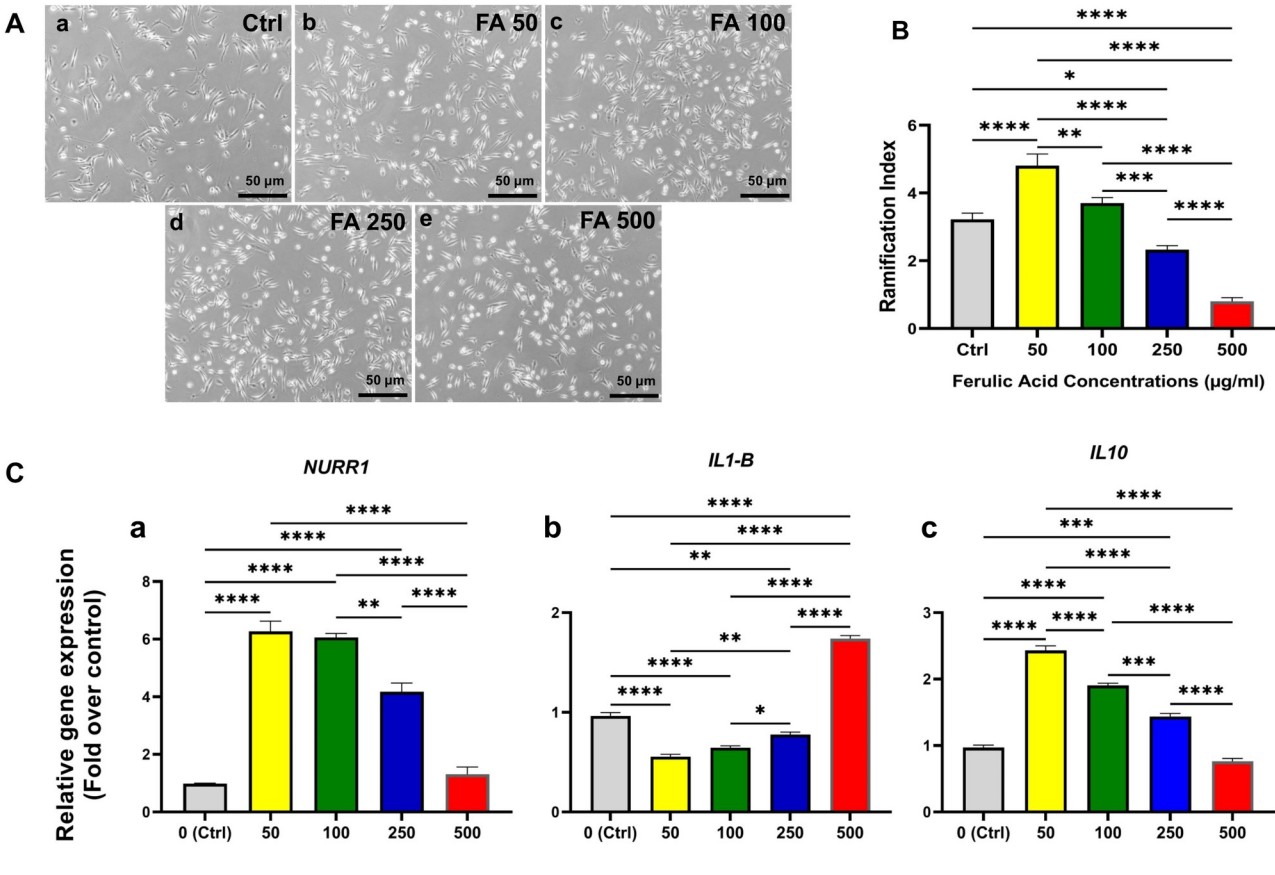

**Fig 2.** Effect of FA on morphology (A), ramification index (B) and gene expressions in cells after treatment with different concentrations of FA. Primary MGCs were treated with different concentrations of FA (50, 100, 250, and 500 μg/ml). As represented in the schematic timeline (Fig 1), cells were treated for 24 hours with 0, 50, 100, 250 and 500 μg/mL of FA (Aa-e). Ramification indexes (RI) from different groups were measured by Sholl analysis tool in Image J software (B). Real-time qPCR was performed to evaluate the expressions of *Nurr1*, *IL1-β* and *IL-10* (C). Scale bars represent 50 μm. Data are presented as mean ± S.E.M, (*: *P < 0.05*, **: *P < 0.01*, ***: *P < 0.001* and ****: *P < 0.0001*).

Based on these experiments, the 50μg/ml FA showed the best results on induction of Nurr1 and IL-10 expressions, so further assessments were performed using this concentration.

## Effect of FA treatment on microglial cells under Aβ stress

As shown in Fig 3Ab, treatment with Aβ strongly affected the morphology of microglial cells, and priming with FA 50 μg/ml could prevent microglial cells from going toward amoeboid form by keeping them in the ramified state as shown in Fig 3Ad. Calculations of RI (Fig 3B) by image j software indicated same results about ramification rate of the treated cells, meaning that FA treatment could preserve RI in FA+ Aβ group.

Also, as shown in Fig 3C, counting the number of amoeboid, rod-like and ramified microglial cells in treated groups showed that the percentage of ramified phenotype was significantly higher in the FA-treated groups (FA and FA+Aβ) than the control and Aβ groups (*P < 0.001*). In the case of the amoeboid phenotype, the highest level was in the Aβ group (*P < 0.0001*). Moreover, about rod like-cells, they were significantly higher in the control and Aβ groups.

According to Fig 3D, assays on gene expressions revealed that the expression levels of *NURR1*, *IL1-β* and *IL10* were greatly changed in groups after the exertion of treatments. The

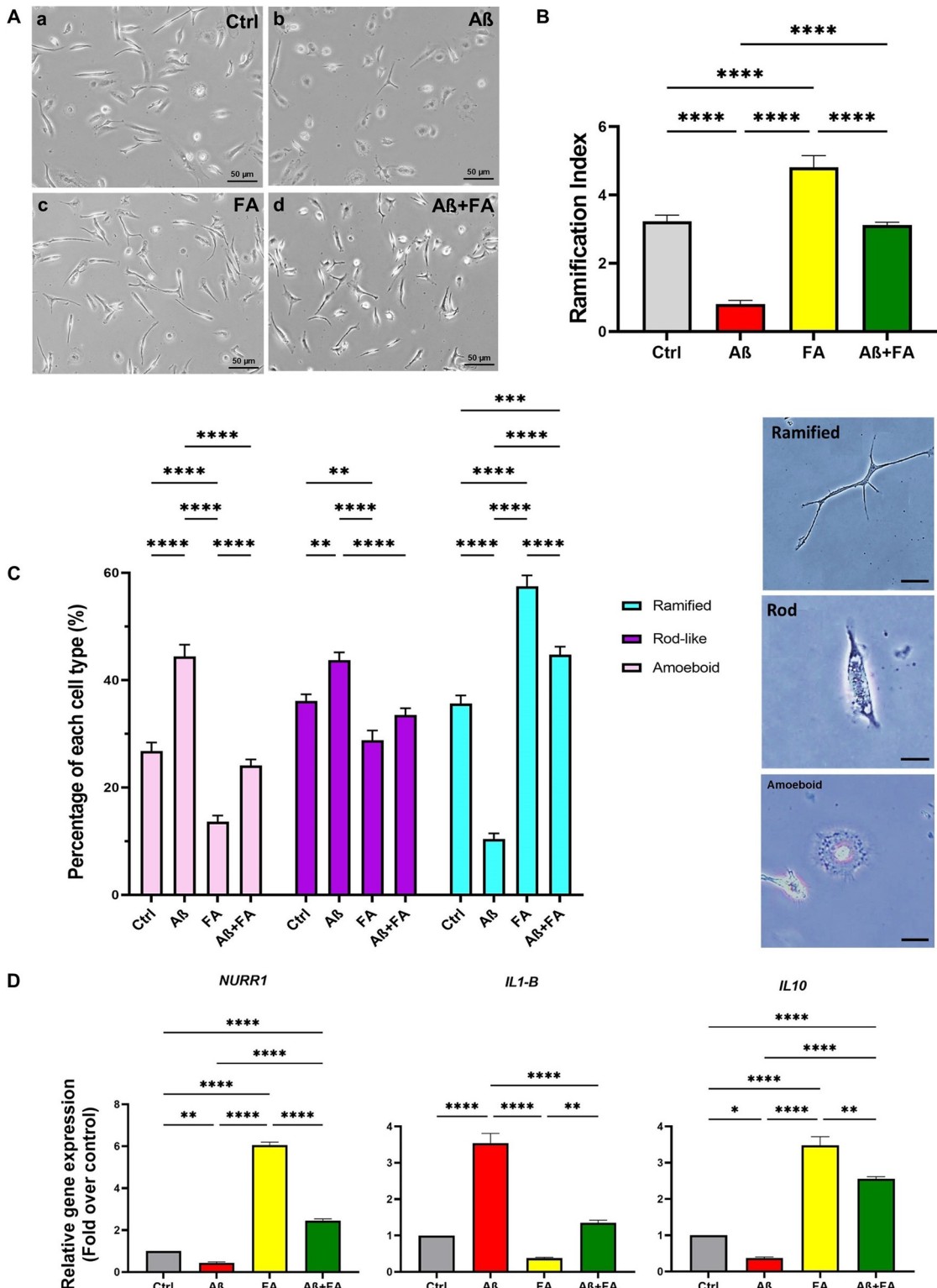

**Fig 3. Effect of FA on MGCs under Aβ-stress.** Changes in cell morphology (A), ramification index (B), percentage of different MGC phenotypes (C), and levels of gene expressions for *Nurr1*, *IL1-β* and *IL-10* (D) were evaluated. Scale bars represent 50 μm. Data are presented as mean ± S.E.M, (*: $P < 0.05$, **: $P < 0.01$, ***: $P < 0.001$ and ****: $P < 0.0001$).

*Nurr1* and *IL10* expressions dramatically declined in the Aβ group compared to the FA and FA+Aβ groups *(P < 0.0001)*. Whereas regarding the *IL1-β* expression the highest level was found in the Aβ group, which was significantly higher than that in all other groups *(P < 0.0001)*. These data revealed that FA treatment could significantly suppress the expression of inflammatory genes while increasing the anti-inflammatory ones.

## Number of Nurr1 positive MGCs

As shown in Fig 4, immunostaining against the Nurr1 protein confirmed the gene expression results and showed that the number of Nurr1 positive cells were highest in the FA treated groups compared to the Aβ group (P < 0.0001). This indicates that the Nurr1 is a valuable marker for the identification of M2 microglial cells, as 29.5% of cells in the FA treated group was Nurr1 positive and almost 13% were positive in the FA+Aβ group. While in the Aβ group, there was a sharp decline in the number of Nurr1 positive cells (2.7%).

## Discussion

In most of the studies conducted so far on the role of Nurr1 in the nervous system, the main focus were on studying its role in dopaminergic differentiation and protection of these neurons against damaging stresses; consequently, limited knowledge has been obtained about its role in microglia and the inflammatory system of the brain [43–46].

This study investigated the effect of FA on the expression of the Nurr1 marker in primary MGCs extracted from the brain of male neonate mice. These cells were treated with an adequate concentration of FA (50 μg/ml), and this treatment affected the morphology, RI, and gene expressions in these cells.

Our results showed that FA could induce the Nurr1 expression, inhibiting the beta-amyloid (Aβ) induced inflammatory reactions in MGCs (activated MGCs). This effect was also evident in the morphological changes, as the Aβ-stressed MGCs that were transformed into an amoeboid form were remodeled into a branched form after treatment with FA, which was also reflected in the RI calculations of MGCs.

In order to induce inflammation, Aβ stress was employed. The findings of this section revealed that treating MGCs with Aβ resulted in an increase in the expression of the pro-inflammatory cytokine IL-1β, while the anti-inflammatory cytokines IL-10 and Nurr1 were decreased. These changes in cytokine expression are believed to be responsible for the shift from ramified to amoeboid. Immunostaining findings also showed that fewer Nurr1 positive MGCs could be detected under Aβ stress. Therefore, these findings confirm that a reduction in Nurr1 activity can increase the inflammatory responses in microglial cells.

Our study and most of the studies that have been conducted on the relationship between Nurr1 and inflammation in MGCs have agreed that the intensity of inflammation has a reverse relationship with the level of Nurr1 expression [32, 47]. Although some studies have reported conflicting reports in this regard, for example, Lallier et al. reported that the role of Nurr1 in controlling LPS-induced brain inflammation is not highly dependent on MGCs [26]. They reported that applying hyperoxia to BV2 cells (immortal MG Cell line) stimulates the expression of Nurr1, but combining it with LPS reduces the intensity of the increased expression of Nurr1 in these cells. However, their study still shows that the level of Nurr1 expression has an inverse correlation with the inflammatory state in MGCs [26]. Overall, comparing the results of our research with other researchers regarding the relationship between the intensity of inflammation and Nurr1 expression, the results of most studies are consistent with our results and prove that MGCs under LPS or Aβ stress show lower levels of Nurr1 expression [29, 48].

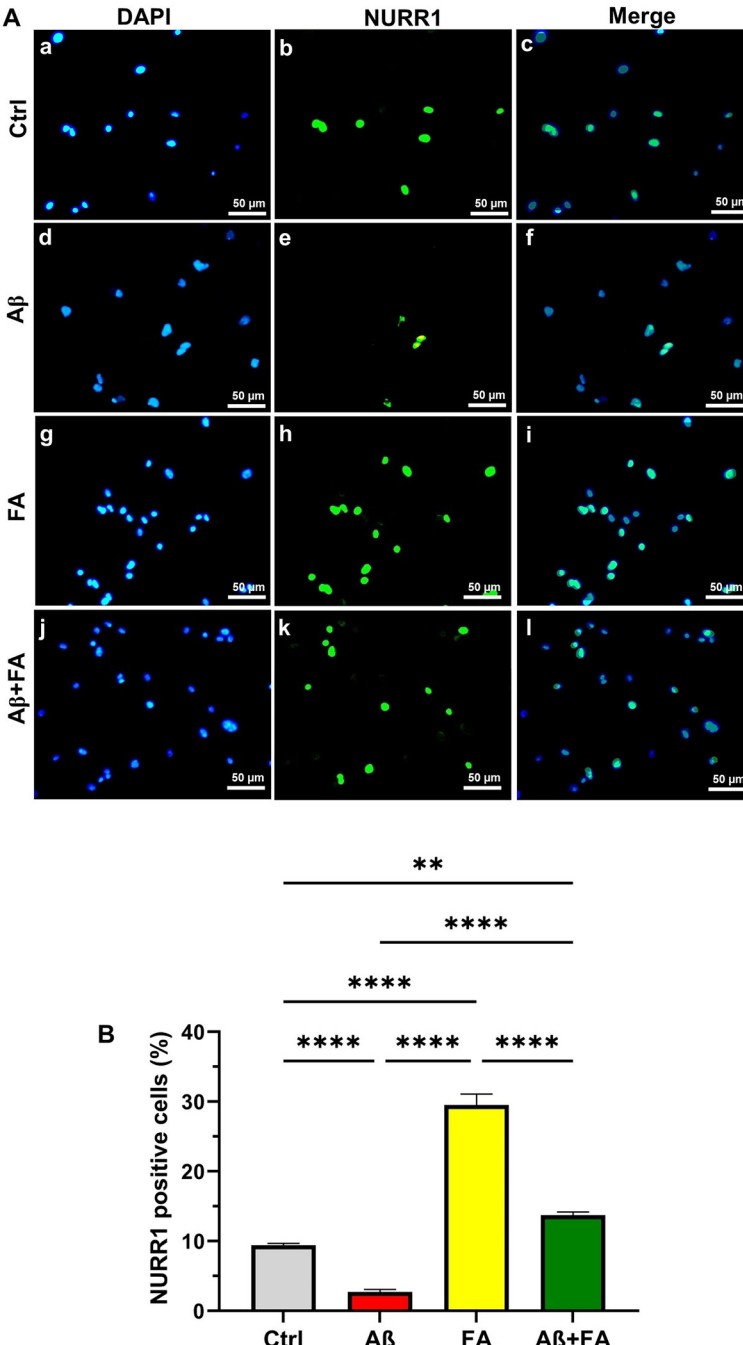

**Fig 4. Immunostaining against Nurr1 protein in MGCs.** The number of Nurr1 positive cells were counted in images captured from each group and their differences were analyzed. The Green color shows the Nurr1 and the blue color denotes the nuclei of cells stained with DAPI. Scale bars represent 50 μm. Data are presented as mean ± S.E.M, (**: *P < 0.01* and ****: *P < 0.0001*).

Also, it has been repeatedly reported that stimulation of MGCs with Aβ oligomers leads to increased expression of pro-inflammatory cytokines such as IL-6, IL-1β, and TNF-α [49–51].

For instance, Kim et al. investigated the inhibitory effect of donepezil on MGCs under Aβ-stress *in vitro* and *in vivo*, and they found that Aβ-stress elevates the expressions of pro-inflammatory signals such as IL-1β, TNF-α, and NO [52]. Similarly, Chen et al. found similar results

when investigating the protective effect of Mogrol against memory disorders caused by Aβ [53].

Also, studies have been conducted on the change of morphology of microglia, and it shows that microglia are branched in a resting state, and after activation, they lose their appendages and become amoeboid [54].

FA is one of the medicinal compounds whose anti-inflammatory effects on the nervous system have been investigated in several studies [2, 7, 12, 55]. Our study showed that the treatment of Aβ-stressed MGCs with FA could significantly increase the expression of Nurr1 compared to the untreated group. In terms of morphology, the MGCs had a branched form, and a less amoebic form was seen, and this result was also proved by quantitative measurement of their RI. Immuno-staining also confirmed that the number of Nurr1 positive MGCs was higher than in the untreated group. Analysis of gene expression also showed that 24-hour treatment with FA could decrease the expression of pro-inflammatory cytokine IL1-β and increase the expression of anti-inflammatory cytokine IL-10.

Studies on animal models have also shown that brain inflammation is inhibited in animals that have received FA due to the reductions in the expression of pro-inflammatory cytokines IL-6, IL-1β, and TNF-α [56]. *In vitro* studies also confirmed the same results indicating that the expression of pro-inflammatory genes involved in initiating LPS-induced inflammation in MGCs was reduced after treatment with FA [55, 57].

Other studies on Aβ1-42-treated rodent microglia and the therapeutic effect of potential herbal medicines for the control of neuro-inflammatory signals have reported that active components of herbal medicines act through reduction of microglial inflammation by inhibiting the inflammatory cytokines such as TNF-α, IL-1β and IL-6 and chemokines (CXCL1 and CCL-2), NF-κB and VEGF/Flt-1 signaling pathways [58, 59].

The inhibition of NLRP3 inflammasome, a significant contributor to Aβ-induced inflammation, is one of the potent mechanisms that reflect the anti-neuroinflammatory function of Nurr1 [59]. There are studies indicating the role of Nurr1 deficiency and the activation of NLRP3 inflammasome axis that can affect the survival rate of dopaminergic neurons [60] and activation of Muller glial cells [61]. Therefore, the inhibition of this axis in microglial cells is one of the potential mechanisms of FA function.

High-mobility group box 1 (HMGB1) is another important damage-associated molecular pattern produced by microglia and astrocytes which can activate NLRP3 inflammasome axis [62], and there are evidences that FA can suppresses the HMGB1 in some types of cells such as hepatocytes [63].

Vasculature alterations are very important pathological findings of Parkinson disease that promote the production of inflammasomes [29], and some evidences suggest that FA can protect brain vasculature by regulating the neurovascular unit [64]. We recommend complementary studies on these pathways to find the possible mechanisms of FA against neuroinflammation.

Beside activation in response to inflammatory signals aging process by itself can affect MGCs, it has been reported that aged microglia display a more reactive/activated phenotype in both human and rat brains [65]. Furthermore, compelling evidence suggests that the expression of NURR1 is slightly down-regulated during the progression of Parkinson's disease [29]. Therefore, the development of effective therapeutics targeting NURR1 pathway holds promise as a preventive approach for neurodegenerative disorders. And further studies need to be done to evaluate the effect of FA on normal aging process and microglial activation.

## Conclusion

The findings of this study suggest that FA is a promising drug or supplementary candidate for inhibiting neuroinflammation caused by Aβ-reactive microglia. The anti-inflammatory effect

of FA is attributed, in part, to its ability to upregulate the expression of the transcription factor Nurr1 in stressed microglia. Furthermore, FA was found to increase Nurr1 and IL-10 expression in homeostatic microglia, making it a potent candidate for preventing AD and age-related microglial activations. Further complementary studies are recommended to investigate the mechanism of FA action on the NLRP3 inflammasome axis in microglial cells and the neuro-vascular unit.

## Supporting information

**S1 Dataset. The complete raw data (minimal data set) for all graphs in this manuscript can be found in the article's supplementary data (S1_minimal_dataset).**
(PDF)

## Acknowledgments

The authors would like to thank the staff of the Royan Institute for biotechnology for their support.

## Author Contributions

**Conceptualization:** Farshad Homayouni Moghadam, Mohammad Hossein Nasr-Esfahani.

**Data curation:** Ali Moghimi-Khorasgani.

**Formal analysis:** Ali Moghimi-Khorasgani.

**Investigation:** Ali Moghimi-Khorasgani.

**Methodology:** Ali Moghimi-Khorasgani.

**Project administration:** Farshad Homayouni Moghadam, Mohammad Hossein Nasr-Esfahani.

**Software:** Ali Moghimi-Khorasgani.

**Supervision:** Farshad Homayouni Moghadam, Mohammad Hossein Nasr-Esfahani.

**Validation:** Ali Moghimi-Khorasgani, Farshad Homayouni Moghadam.

**Visualization:** Ali Moghimi-Khorasgani.

**Writing – original draft:** Ali Moghimi-Khorasgani, Farshad Homayouni Moghadam.

**Writing – review & editing:** Farshad Homayouni Moghadam, Mohammad Hossein Nasr-Esfahani.

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
