## [Decision Letter · Decision Letter 0]

20 Jun 2023

PONE-D-23-15118Ferulic Acid reduces amyloid beta mediated neuroinflammation through modulation of Nurr1 expression in microglial cellsPLOS ONE

Dear Dr. Homayouni Moghadam,

Thank you for submitting your manuscript to PLOS ONE. After careful consideration, we feel that it has merit but does not fully meet PLOS ONE’s publication criteria as it currently stands. Therefore, we invite you to submit a revised version of the manuscript that addresses the points raised during the review process.

We look forward to receiving your revised manuscript.

Kind regards,

Weidong Le

Academic Editor

PLOS ONE

Journal Requirements:

Reviewers' comments:

Reviewer's Responses to Questions

**Comments to the Author**

1. Is the manuscript technically sound, and do the data support the conclusions?

Reviewer #1: Yes

Reviewer #2: Yes

2. Has the statistical analysis been performed appropriately and rigorously? 

Reviewer #1: Yes

Reviewer #2: Yes

3. Have the authors made all data underlying the findings in their manuscript fully available?

Reviewer #1: Yes

Reviewer #2: Yes

4. Is the manuscript presented in an intelligible fashion and written in standard English?

Reviewer #1: Yes

Reviewer #2: Yes

5. Review Comments to the Author

Reviewer #1: This article focuses on the function of Ferulic acid, a natural phenolic compound, on neuroinflammation in microglial cells. In vitro with beta-amyloid (Aβ) intervention, the authors found that FA could restore the levels of IL-10 and Nurr1 while reduce the level of IL1-β in microglial cells, and increase the ramification index of microglial cells and the number of NURR1 positive cells. The results suggested that FA may be a candidate drug for anti-microglia proliferation in nervous system.

The research is clear, with high clinical significance. The language is concise and easy to understand. Although I appreciate the author' effort on this paper, but the discussion needs to be more in-depth. So I suggest that the authors may want to cite and discuss some recent findings as follows in the discussion to improve the article：

1. Dl-3-n-Butylphthalide Rescues Dopaminergic Neurons in Parkinson’s Disease Models by Inhibiting the NLRP3 Inflammasome and Ameliorating Mitochondrial Impairment. Front Immunol. 2021 Dec 1;12:794770.

2. Jeon, S. G., Song, E. J., Lee, D., Park, J., Nam, Y., Kim, J. I., & Moon, M. (2019). Traditional Oriental Medicines and Alzheimer's Disease. Aging and disease, 10(2), 307–328. https://doi.org/10.14336/AD.2018.0328

3. The link between neuroinflammation and the neurovascular unit in synucleinopathies. Science Advances, 2023, 9(7), eabq1141.

4. Yuan, M., Wang, Y., Wang, S., Huang, Z., Jin, F., Zou, Q., Li, J., Pu, Y., & Cai, Z. (2021). Bioenergetic Impairment in the Neuro-Glia-Vascular Unit: An Emerging Physiopathology during Aging. Aging and disease, 12(8), 2080–2095. https://doi.org/10.14336/AD.2021.04017.

5. Causal effect of gut-microbiota-derived metabolite trimethylamine N-oxide on Parkinson's disease: A Mendelian randomization study. European journal of neurology, 10.1111/ene.15702.

6. Abubakar, M. B., Sanusi, K. O., Ugusman, A., Mohamed, W., Kamal, H., Ibrahim, N. H., Khoo, C. S., & Kumar, J. (2022). Alzheimer's Disease: An Update and Insights Into Pathophysiology. Frontiers in aging neuroscience, 14, 742408. https://doi.org/10.3389/fnagi.2022.742408

7. Multi-modal analysis of gene expression from postmortem brains and blood identifies synaptic vesicle trafficking genes to be associated with Parkinson’s disease. Briefings in Bioinformatics 2020 Oct 20: bbaa244. doi: 10.1093/bib/bbaa244.

8. Agrawal, I., & Jha, S. (2020). Mitochondrial Dysfunction and Alzheimer's Disease: Role of Microglia. Frontiers in aging neuroscience, 12, 252. https://doi.org/10.3389/fnagi.2020.00252.

Reviewer #2: Regarding the manuscript PONE-D-23-15118 titled: "Ferulic Acid reduces amyloid beta mediated Neuroinflammation through Modulation of Nurr1 Expression in Microglial Cells.” Neuroinflammation plays a crucial role in the progression of neurodegenerative disorders. Glial cell activation and subsequent adaptive immune involvement are the main neuroinflammatory features discussed in the submitted article. The authors studied the potential role of ferulic acid in modulating the transition of microglia to reactive states due to its anti-inflammatory properties and ability to induce Nurr1 expression.

Nurr1 and neuroinflammation have already been extensively studied in previous research. While the specific focus on the effects of ferulic acid on Nurr1 expression in microglial cells may be a novel aspect of the study, it is likely building upon existing knowledge in the field. However, the current version of the manuscript still needs revision before being ready for publication.

My evaluation of the paper is as follows:

Strong Points:

- The study investigates the effect of ferulic acid treatment on microglial cells and its potential role in modulating Nurr1 expression, which has implications for neuroinflammation.

- The study provides valuable insights into the morphological changes and gene expression alterations induced by ferulic acid treatment, highlighting its potential anti-inflammatory properties in microglial cells.

- The study lays the foundation for further research by identifying ferulic acid as a potential therapeutic candidate for targeting microglial activation and neuroinflammatory processes, offering new avenues for developing interventions.

Points to Improve:

The following suggestions may help the authors improve the manuscript.

- Some sentences are lengthy and complex, making the content difficult to understand. I recommend breaking them down into smaller, more concise sentences to enhance clarity.

- Please double-check the manuscript for any linguistic typos and correct them accordingly. I have corrected some, as will be mentioned below.

- Ensure consistency in using abbreviations throughout the manuscript, per the journal guidelines. Ensure there were defined in the first instance" not to mention the abbreviation and full name many times in every section; please check the journal guideline time, for example (ramification index (RI), beta-amyloid (Aβ) ………. If an abbreviation has been defined once, it is unnecessary to repeatedly add it unless required for clarity, as α-synucleinopathy (αS-pathy) is unnecessary.

- The terminology used to describe the microglial cells, such as "reactive phenotype (amoeboid, classical activated; cytotoxic, aggressive; phagocytic; M1)" and "alternative activated M2 type (anti-inflammatory; neuroprotective)." The terminology is not entirely accurate or consistent, and these are expected, but recently trying to unify the nomenclature. It would be more appropriate to update and cite the nomenclature of microglia inflammation states, as per the updated nomenclature: *_"Neuron. 2022 Nov 2; 110(21):3458–3483. doi: 10.1016/j.neuron.2022.10.020. Microglia states and nomenclature: A field at its crossroads.” And the recent comprehensive review on the role of Nurr1 in neuroinflammation*_” Advances in NURR1-Regulated Neuroinflammation Associated with Parkinson’s Disease: https://www.mdpi.com/1422-0067/23/24/16184 doi: 10.3390/ijms232416184.”

- The abstract briefly outlines the study's objectives, methods, and key findings. However, it does not adequately summarize the main results and conclusions, nor does it mention the significance or implications of the findings in the broader context of the field. Please revise the abstract accordingly.

- Mechanistic insights: While the study explores the effects of FA on microglial cells and the induction of Nurr1 expression, it would be beneficial to include more mechanistic insights into how FA mediates these effects. Discussion of the potential signaling pathways or molecular mechanisms through which FA influences Nurr1 expression and modulates microglial responses would enhance the understanding of FA's anti-inflammatory properties.

- As approved, Ferulic acid inhibits inflammation, and cytokine secretion is one of the assessments for anti-inflammatory properties. It will be perfect for discussing your results with previous studies (https://pubmed.ncbi.nlm.nih.gov/27532877/ and https://www.ncbi.nlm.nih.gov/pmc/articles/PMC6281882/), would strengthen the study's findings and demonstrate the functional implications of FA treatment on microglial behavior.

- Limited discussion of potential limitations: The discussion section does not adequately address the study's potential limitations. For instance, the study only used an in vitro model of microglial. While the in vitro model used in this study provides valuable insights, validating the findings in an in vivo model would be important. The author needs to highlight future perspectives and limitations of their work, especially the animal models, or utilize other relevant in vivo approaches to confirm the observed effects of FA on microglial cells and provide a more accurate representation of its potential therapeutic benefits.

- In line 85, "α-synucleinopathy (αS-pathy)" is mentioned as a condition caused by metabolic changes in dopaminergic neurons associated with reduced Nurr1 expression. However, α-synucleinopathy is primarily associated with PD and α-synuclein aggregation. Please revise it to “α-synucleinopathy is often associated with PD. Dysfunction or reduced expression of Nurr1 has been associated with impaired energy metabolism, mitochondrial dysfunction, and oxidative stress, all implicated in PD pathogenesis.”

- In line 116, it is mentioned that studies on murine primary microglia are more reliable than studies on routine human and murine cell lines. However, it should be clarified that murine primary microglia are more relevant for studying microglial function in the murine system but may not fully represent human microglia. Please double-check.

- In line 139, "derbies" should be corrected to "debris."

- In line 188, it should be clarified that Nurr1 and DAPI-stained cells and nuclei were counted separately to determine the number of Nurr1-positive cells.

- The conclusion should clearly state the study's overall conclusions and the results' significance and briefly mention potential future directions.

6. PLOS authors have the option to publish the peer review history of their article (what does this mean?). If published, this will include your full peer review and any attached files.

Reviewer #1: No

Reviewer #2: No

---

## [Author Response · Author response to Decision Letter 0]

16 Jul 2023

Dear Editor

Thank you very much for considering our manuscript (PONE-D-23-15118) for publication in your valuable journal. Typewriting errors were fixed according to the journal MANUSCRIPT BODY FORMATTING GUIDELINES.

We provided the minimal dataset of our manuscript as an PDF file containing the all raw data for all graphs in separate sheets and is uploaded as a supporting Information file S1_minimal_dataset.

Our answers to the comments of reviewers are as below. 

Dear reviewers

Thank you very much for your detailed and advantageous comments which will definitely add value to our article. Below is the list of comments and amendments made for each of them in order. Also, a track changed version has been uploaded after applying the changes to the article.

Regards

Comments and Answers:

Reviewer #1: 

Comment: The research is clear, with high clinical significance. The language is concise and easy to understand. Although I appreciate the author' effort on this paper, but the discussion needs to be more in-depth. So I suggest that the authors may want to cite and discuss some recent findings as follows in the discussion to improve the article：

1. Dl-3-n-Butylphthalide Rescues Dopaminergic Neurons in Parkinson’s Disease Models by Inhibiting the NLRP3 Inflammasome and Ameliorating Mitochondrial Impairment. Front Immunol. 2021 Dec 1;12:794770.

2. Jeon, S. G., Song, E. J., Lee, D., Park, J., Nam, Y., Kim, J. I., & Moon, M. (2019). Traditional Oriental Medicines and Alzheimer's Disease. Aging and disease, 10(2), 307–328. https://doi.org/10.14336/AD.2018.0328

3. The link between neuroinflammation and the neurovascular unit in synucleinopathies. Science Advances, 2023, 9(7), eabq1141.

4. Yuan, M., Wang, Y., Wang, S., Huang, Z., Jin, F., Zou, Q., Li, J., Pu, Y., & Cai, Z. (2021). Bioenergetic Impairment in the Neuro-Glia-Vascular Unit: An Emerging Physiopathology during Aging. Aging and disease, 12(8), 2080–2095. https://doi.org/10.14336/AD.2021.04017.

5. Causal effect of gut-microbiota-derived metabolite trimethylamine N-oxide on Parkinson's disease: A Mendelian randomization study. European journal of neurology, 10.1111/ene.15702.

6. Abubakar, M. B., Sanusi, K. O., Ugusman, A., Mohamed, W., Kamal, H., Ibrahim, N. H., Khoo, C. S., & Kumar, J. (2022). Alzheimer's Disease: An Update and Insights Into Pathophysiology. Frontiers in aging neuroscience, 14, 742408. https://doi.org/10.3389/fnagi.2022.742408

7. Multi-modal analysis of gene expression from postmortem brains and blood identifies synaptic vesicle trafficking genes to be associated with Parkinson’s disease. Briefings in Bioinformatics 2020 Oct 20: bbaa244. doi: 10.1093/bib/bbaa244.

8. Agrawal, I., & Jha, S. (2020). Mitochondrial Dysfunction and Alzheimer's Disease: Role of Microglia. Frontiers in aging neuroscience, 12, 252. https://doi.org/10.3389/fnagi.2020.00252.

Answer: all of these articles were used to revise and to add paragraphs to the discussion section.

 “Other studies on Aβ1-42-treated rodent microglia and the therapeutic effect of potential herbal medicines for the control of neuro-inflammatory signals have reported that active components of herbal medicines act through reduction of microglial inflammation by inhibiting the inflammatory cytokines such as TNF-α, IL-1β and IL-6 and chemokines (CXCL1 and CCL-2), NF-κB and VEGF/Flt-1 signaling pathways [54, 55].

The inhibition of NLRP3 inflammasome, a significant contributor to Aβ-induced inflammation, is one of the potent mechanisms that reflect the anti-neuroinflammatory function of Nurr1[55]. There are studies indicating the role of Nurr1 deficiency and the activation of NLRP3 inflammasome axis that can affect the survival rate of dopaminergic neurons [56] and activation of Muller glial cells [57]. Therefore, the inhibition of this axis in microglial cells is one of the potential mechanisms of FA function.

High-mobility group box 1 (HMGB1) is another important damage-associated molecular pattern produced by microglia and astrocytes which can activate NLRP3 inflammasome axis [58], and there are evidences that FA can suppresses the HMGB1 in some types of cells such as hepatocytes [59].

Vasculature alterations are very important pathological findings of Parkinson disease that promote the production of inflammasomes [60], and some evidences suggest that FA can protect brain vasculature by regulating the neurovascular unit [61]. We recommend complementary studies on these pathways to find the possible mechanisms of FA against neuroinflammation. 

Beside activation in response to inflammatory signals aging process by itself can affect MGCs, it has been reported that aged microglia display a more reactive/activated phenotype in both human and rat brains [62]. Furthermore, compelling evidence suggests that the expression of NURR1 is slightly down-regulated during the progression of Parkinson's disease [60].”

Reviewer #2: 

Comment 1: Some sentences are lengthy and complex, making the content difficult to understand. I recommend breaking them down into smaller, more concise sentences to enhance clarity.

Answer: lengthy and complex sentences were simplified. 

“Research on microglial cells has revealed that FA exhibits potent anti-oxidative and anti-inflammatory properties, effectively reducing the release of pro-inflammatory cytokines while increasing the expression of anti-inflammatory cytokines. [12]. In vivo studies have shown that treatment with FA reduces inflammation by reducing IL-1β expression in the models of neurodegenerative disorders [7, 13, 14].”

“To date, no studies have investigated the effect of FA on Nurr1 expression in primary microglial cells. However, the expression of Nurr1 increases in neurons treated with FA, which leads to the expression of dopaminergic differentiation markers [32]. According to the findings proving the anti-inflammatory effect of FA and the role of Nurr1 in neuroinflammation, the ability of FA to influence the expression of Nurr1 in mouse primary microglia was investigated in the present study.”

Comment: - Please double-check the manuscript for any linguistic typos and correct them accordingly. I have corrected some, as will be mentioned below.

- Ensure consistency in using abbreviations throughout the manuscript, per the journal guidelines. Ensure there were defined in the first instance" not to mention the abbreviation and full name many times in every section; please check the journal guideline time, for example (ramification index (RI), beta-amyloid (Aβ) ………. If an abbreviation has been defined once, it is unnecessary to repeatedly add it unless required for clarity, as α-synucleinopathy (αS-pathy) is unnecessary.

Answer: abbreviations were fixed according to the comment and unnecessary abbreviations were deleted.

Comment: - The terminology used to describe the microglial cells, such as "reactive phenotype (amoeboid, classical activated; cytotoxic, aggressive; phagocytic; M1)" and "alternative activated M2 type (anti-inflammatory; neuroprotective)." The terminology is not entirely accurate or consistent, and these are expected, but recently trying to unify the nomenclature. It would be more appropriate to update and cite the nomenclature of microglia inflammation states, as per the updated nomenclature: *_"Neuron. 2022 Nov 2; 110(21):3458–3483. doi: 10.1016/j.neuron.2022.10.020. Microglia states and nomenclature: A field at its crossroads.” And the recent comprehensive review on the role of Nurr1 in neuroinflammation*_” Advances in NURR1-Regulated Neuroinflammation Associated with Parkinson’s Disease: https://www.mdpi.com/1422-0067/23/24/16184 doi: 10.3390/ijms232416184.”

Answer: the terminology for naming the cell types were unified and above articles were cited to provide the data for readers. For this comments we checked all of the document to use similar terminology in the whole text and many changes were made so please review whole of the document. some of them are as below:

“With various phenotypes, they can shift from a homeostatic state to a reactive state or transit from a reactive to a non-inflammatory reactive state (alternative homeostatic). A well-timed transit is crucial in limiting excessive microglial reaction and promoting the healing process.”

“Morphological assessments and measurements of the RI revealed that FA treatment reversed amoeboid and rod-like cells to a ramified state, which is specific morphology for non-inflammatory reactive microglia.”

“To conclude, FA can provide potential alternative homeostatic transition in Aβ-reactive microglia by recruiting the NURR1 dependent anti-inflammatory responses. This makes it a promising therapeutic candidate for suppressing Aβ-induced neuroinflammatory responses in MGCs. Furthermore, given that FA has the ability to increase NURR1 levels in homeostatic microglia, it could be utilized as a preventative medication.”

- The abstract briefly outlines the study's objectives, methods, and key findings. However, it does not adequately summarize the main results and conclusions, nor does it mention the significance or implications of the findings in the broader context of the field. Please revise the abstract accordingly.

Answer: main results and conclusions were summarized and mentioned in the abstract with more detail. 

“Treating MGCs with FA (50 μg/ml) induced Nurr1 and IL-10 expressions, while reducing the level of IL-1β in the absence of Aβ-stress. Further assessments on cells under Aβ-stress showed that FA treatment restored the IL-10 and Nurr1 levels, increased the RI of cells, and the number of NURR1-positive cells. Morphological assessments and measurements of the RI revealed that FA treatment reversed amoeboid and rod-like cells to a ramified state, which is specific morphology for non-inflammatory reactive microglia.

To conclude, FA can provide potential alternative homeostatic transition in Aβ-reactive microglia by recruiting the NURR1 dependent anti-inflammatory responses. This makes it a promising therapeutic candidate for suppressing Aβ-induced neuroinflammatory responses in MGCs. Furthermore, given that FA has the ability to increase NURR1 levels in homeostatic microglia, it could be utilized as a preventative medication.”

Comment: - Mechanistic insights: While the study explores the effects of FA on microglial cells and the induction of Nurr1 expression, it would be beneficial to include more mechanistic insights into how FA mediates these effects. Discussion of the potential signaling pathways or molecular mechanisms through which FA influences Nurr1 expression and modulates microglial responses would enhance the understanding of FA's anti-inflammatory properties.

Answer: candidate signaling pathways or molecular mechanisms behind FA effect on Nurr1 expression were added to the text.

“Other studies on Aβ1-42-treated rodent microglia and the therapeutic effect of potential herbal medicines for the control of neuro-inflammatory signals have reported that active components of herbal medicines act through reduction of microglial inflammation by inhibiting the inflammatory cytokines such as TNF-α, IL-1β and IL-6 and chemokines (CXCL1 and CCL-2), NF-κB and VEGF/Flt-1 signaling pathways [54, 55].

The inhibition of NLRP3 inflammasome, a significant contributor to Aβ-induced inflammation, is one of the potent mechanisms that reflect the anti-neuroinflammatory function of Nurr1[55]. There are studies indicating the role of Nurr1 deficiency and the activation of NLRP3 inflammasome axis that can affect the survival rate of dopaminergic neurons [56] and activation of Muller glial cells [57]. Therefore, the inhibition of this axis in microglial cells is one of the potential mechanisms of FA function.

High-mobility group box 1 (HMGB1) is another important damage-associated molecular pattern produced by microglia and astrocytes which can activate NLRP3 inflammasome axis [58], and there are evidences that FA can suppresses the HMGB1 in some types of cells such as hepatocytes [59].

Vasculature alterations are very important pathological findings of Parkinson disease that promote the production of inflammasomes [60], and some evidences suggest that FA can protect brain vasculature by regulating the neurovascular unit [61]. We recommend complementary studies on these pathways to find the possible mechanisms of FA against neuroinflammation. 

Beside activation in response to inflammatory signals aging process by itself can affect MGCs, it has been reported that aged microglia display a more reactive/activated phenotype in both human and rat brains [62]. Furthermore, compelling evidence suggests that the expression of NURR1 is slightly down-regulated during the progression of Parkinson's disease [60]. Therefore, the development of effective therapeutics targeting NURR1 pathway holds promise as a preventive approach for neurodegenerative disorders. And further studies need to be done to evaluate the effect of FA on normal aging process and microglial activation.”

Comment: - As approved, Ferulic acid inhibits inflammation, and cytokine secretion is one of the assessments for anti-inflammatory properties. It will be perfect for discussing your results with previous studies (https://pubmed.ncbi.nlm.nih.gov/27532877/ and https://www.ncbi.nlm.nih.gov/pmc/articles/PMC6281882/), would strengthen the study's findings and demonstrate the functional implications of FA treatment on microglial behavior.

Answer: thank you for this comment, we had it already and we used this useful reference to add some sentences to our manuscript.

“Lallier et al. (2016) found that while whole-brain homogenates show an increase in Nurr1 expression in response to inflammation, primary microglia and BV2 cells exhibit a decrease in Nurr1 expression under similar conditions. These findings suggest that Nurr1 expression may increase in other cell types rather than microglia [26].

Although some studies have reported conflicting reports in this regard, for example, Lallier et al. reported that the role of Nurr1 in controlling LPS-induced brain inflammation is not highly dependent on MGCs [26]. They reported that applying hyperoxia to BV2 cells (immortal MG Cell line) stimulates the expression of Nurr1, but combining it with LPS reduces the intensity of the increased expression of Nurr1 in these cells. However, their study still shows that the level of Nurr1 expression has an inverse correlation with the inflammatory state in MGCs [26].”

Comment: - Limited discussion of potential limitations: The discussion section does not adequately address the study's potential limitations. For instance, the study only used an in vitro model of microglial. While the in vitro model used in this study provides valuable insights, validating the findings in an in vivo model would be important. The author needs to highlight future perspectives and limitations of their work, especially the animal models, or utilize other relevant in vivo approaches to confirm the observed effects of FA on microglial cells and provide a more accurate representation of its potential therapeutic benefits.

Answer: potential limitations and future perspective were added to the manuscript, and more relevant in vivo approaches were added to the text.

“Studies on animal models have also shown that brain inflammation is inhibited in animals that have received FA due to the reductions in the expression of pro-inflammatory cytokines IL-6, IL-1β, and TNF-α [52]. In vitro studies also confirmed the same results indicating that the expression of pro-inflammatory genes involved in initiating LPS-induced inflammation in MGCs was reduced after treatment with FA [51, 53].

Other studies on Aβ1-42-treated rodent microglia and the therapeutic effect of potential herbal medicines for the control of neuro-inflammatory signals have reported that active components of herbal medicines act through reduction of microglial inflammation by inhibiting the inflammatory cytokines such as TNF-α, IL-1β and IL-6 and chemokines (CXCL1 and CCL-2), NF-κB and VEGF/Flt-1 signaling pathways [54, 55].

The inhibition of NLRP3 inflammasome, a significant contributor to Aβ-induced inflammation, is one of the potent mechanisms that reflect the anti-neuroinflammatory function of Nurr1[55]. There are studies indicating the role of Nurr1 deficiency and the activation of NLRP3 inflammasome axis that can affect the survival rate of dopaminergic neurons [56] and activation of Muller glial cells [57]. Therefore, the inhibition of this axis in microglial cells is one of the potential mechanisms of FA function.”

Comment: - In line 85, "α-synucleinopathy (αS-pathy)" is mentioned as a condition caused by metabolic changes in dopaminergic neurons associated with reduced Nurr1 expression. However, α-synucleinopathy is primarily associated with PD and α-synuclein aggregation. Please revise it to “α-synucleinopathy is often associated with PD. Dysfunction or reduced expression of Nurr1 has been associated with impaired energy metabolism, mitochondrial dysfunction, and oxidative stress, all implicated in PD pathogenesis.”

Answer: we revised them to: 

” Dysfunction or reduced expression of Nurr1 has been associated with impaired energy metabolism, mitochondrial dysfunction, and oxidative stress. Given the association between reduced Nurr1 expression and these changes in dopaminergic neurons, beside anti-synucleinopathy therapy, increasing Nurr1 expression has been suggested as a promising therapeutic target for Parkinson disease.”

Comment: - In line 116, it is mentioned that studies on murine primary microglia are more reliable than studies on routine human and murine cell lines. However, it should be clarified that murine primary microglia are more relevant for studying microglial function in the murine system but may not fully represent human microglia. Please double-check.

Answer: we checked this and we changed it according to the comment.

“We used primary mice microglia as recent studies have shown that immortal microglia cell lines are both genetically and functionally distinct from primary microglia [33]. The presence of microglial states observed in mice and their existence in humans are still a topic of debate. It is essential to validate and translate these findings across species to prevent clinical trial failures resulting from limitations of animal models [34]. Furthermore, since human primary microglia are not easily accessible for research, studying primary murine cell lines could be advantageous in identifying possible similarities and differences.”

Comment: - In line 139, "derbies" should be corrected to "debris."

- In line 188, it should be clarified that Nurr1 and DAPI-stained cells and nuclei were counted separately to determine the number of Nurr1-positive cells. 

Answer: all of the above errors were fixed.

“To determine the number of Nurr1 positive cells, ten images were captured from random areas of each well and Nurr1 and DAPI-stained cells (nuclei) were counted separately for each image.”

Comment: - The conclusion should clearly state the study's overall conclusions and the results' significance and briefly mention potential future directions.

Answer: we checked this and we added the conclusions and future perspectives.

“The findings of this study suggest that FA is a promising drug or supplementary candidate for inhibiting neuroinflammation caused by Aβ-reactive microglia. The anti-inflammatory effect of FA is attributed, in part, to its ability to upregulate the expression of the transcription factor Nurr1 in stressed microglia. Furthermore, FA was found to increase Nurr1 and IL-10 expression in homeostatic microglia, making it a potent candidate for preventing AD and age-related microglial activations. Further complementary studies are recommended to investigate the mechanism of FA action on the NLRP3 inflammasome axis in microglial cells and the neurovascular unit.”

---

## [Decision Letter · Decision Letter 1]

31 Jul 2023

PONE-D-23-15118R1Ferulic Acid reduces amyloid beta mediated neuroinflammation through modulation of Nurr1 expression in microglial cellsPLOS ONE

Dear Dr. Homayouni Moghadam,

Thank you for submitting your manuscript to PLOS ONE. After careful consideration, we feel that it has merit but does not fully meet PLOS ONE’s publication criteria as it currently stands. Therefore, we invite you to submit a revised version of the manuscript that addresses the points raised during the review process.

We look forward to receiving your revised manuscript.

Kind regards,

Weidong Le

Academic Editor

PLOS ONE

Journal Requirements:

**Additional Editor Comments:**

Please double check the references and citations according to the reviewer's comments. 

Reviewers' comments:

Reviewer's Responses to Questions

**Comments to the Author**

1. If the authors have adequately addressed your comments raised in a previous round of review and you feel that this manuscript is now acceptable for publication, you may indicate that here to bypass the “Comments to the Author” section, enter your conflict of interest statement in the “Confidential to Editor” section, and submit your "Accept" recommendation.

Reviewer #1: All comments have been addressed

Reviewer #2: All comments have been addressed

2. Is the manuscript technically sound, and do the data support the conclusions?

Reviewer #1: Yes

Reviewer #2: Partly

3. Has the statistical analysis been performed appropriately and rigorously? 

Reviewer #1: Yes

Reviewer #2: I Don't Know

4. Have the authors made all data underlying the findings in their manuscript fully available?

Reviewer #1: Yes

Reviewer #2: (No Response)

5. Is the manuscript presented in an intelligible fashion and written in standard English?

Reviewer #1: Yes

Reviewer #2: (No Response)

6. Review Comments to the Author

Reviewer #1: The authors have fully addressed my concerns. In addition, the authors have tried their best to answer other reviewers' concerns. It would be acceptance.

Reviewer #2: The author need to double check the citation. Various phrases still missing references and some need to double check the accuracy. For example line 80-89. In the original submission the author mentioned the reference then they deleted in the revision version!!!

7. PLOS authors have the option to publish the peer review history of their article (what does this mean?). If published, this will include your full peer review and any attached files.

Reviewer #1: No

Reviewer #2: No

---

## [Author Response · Author response to Decision Letter 1]

2 Aug 2023

Dear reviewers

Thank you very much for your detailed and advantageous comments which will definitely add value to our article. Below is the list of comments and amendments made for each of them in order. Also, a track changed version has been uploaded after applying the changes to the article.

Regards

Comments and Answers:

Reviewer #1: 

No Comment

Reviewer #2: 

Comment 1: The authors need to double check the citations. Various phrases still missing references and some need to double check the accuracy. For example, line 80-89. In the original submission the author mentioned the reference then they deleted in the revision version!!!

Answer: In the first revision we made a mistake in citations of one of the revised paragraphs and in this second revision we inserted lost citations in that paragraph lines 80-89. Also we added some citations in lines 293-296.

80-89: One of the transcription factors that can inhibit inflammation caused by Aβ is Nurr1 [26, 27], whose structural defects or reduced expression have been shown to be associated with increased amyloid plaque deposition [26]. As a transcription factor, Nurr1 plays a crucial role in the development of dopaminergic neurons and contributes to the activation of neuroprotective and anti-inflammatory pathways [28]. Dysfunction or reduced expression of Nurr1 has been associated with impaired energy metabolism, mitochondrial dysfunction, and oxidative stress [29, 30]. Given the association between reduced Nurr1 expression and these changes in dopaminergic neurons, beside anti-synucleinopathy therapy, increasing Nurr1 expression has been suggested as a promising therapeutic target for Parkinson disease [28]. However, fewer studies have been conducted on its role in microglial and astrocytic inflammations.

293-296: Overall, comparing the results of our research with other researchers regarding the relationship between the intensity of inflammation and Nurr1 expression, the results of most studies are consistent with our results and prove that MGCs under LPS or Aβ stress show lower levels of Nurr1 expression [29, 48].

---

## [Editor Report · Decision Letter 2]

4 Aug 2023

Ferulic Acid reduces amyloid beta mediated neuroinflammation through modulation of Nurr1 expression in microglial cells

PONE-D-23-15118R2

Dear Dr. Homayouni Moghadam,

We’re pleased to inform you that your manuscript has been judged scientifically suitable for publication and will be formally accepted for publication once it meets all outstanding technical requirements.

Kind regards,

Weidong Le

Academic Editor

PLOS ONE
---

## [Editor Report · Acceptance letter]

9 Aug 2023

PONE-D-23-15118R2 

Ferulic Acid reduces amyloid beta mediated neuroinflammation through modulation of Nurr1 expression in microglial cells 

Dear Dr. Homayouni Moghadam:

I'm pleased to inform you that your manuscript has been deemed suitable for publication in PLOS ONE. Congratulations! Your manuscript is now with our production department. 

Kind regards, 

on behalf of

Dr. Weidong Le 

Academic Editor

PLOS ONE